# Enhancing Lettuce (*Lactuca sativa*) Productivity: Foliar Sprayed Fe-Alg-CaCO_3_ MPs as Fertilizers for Aquaponics Cultivation

**DOI:** 10.3390/plants13233416

**Published:** 2024-12-05

**Authors:** Davide Frassine, Roberto Braglia, Francesco Scuderi, Enrico Luigi Redi, Federica Valentini, Michela Relucenti, Irene Angela Colasanti, Andrea Macchia, Ivo Allegrini, Angelo Gismondi, Gabriele Di Marco, Antonella Canini

**Affiliations:** 1Biology Department, Tor Vergata University of Rome, Via della Ricerca Scientifica 1, 00133 Rome, Italy; davide.frassine@students.uniroma2.eu (D.F.); francesco.scuderi@uniroma2.it (F.S.); enrico.luigi.redi@uniroma2.it (E.L.R.); gismondi@scienze.uniroma2.it (A.G.); gabriele.di.marco@uniroma2.it (G.D.M.); canini@uniroma2.it (A.C.); 2PhD Program in Evolutionary Biology and Ecology, Tor Vergata University of Rome, Via della Ricerca Scientifica 1, 00133 Rome, Italy; 3Sciences and Chemical Technologies Department, Tor Vergata University of Rome, Via della Ricerca Scientifica 1, 00133 Rome, Italy; federicavalentini.chem@gmail.com (F.V.); ireneangela.colasanti@students.uniroma2.eu (I.A.C.); 4Department of Anatomy, Histology, Forensic Medicine, and Orthopedics, Sapienza University of Rome, Via Alfonso Borelli 50, 00161 Rome, Italy; michela.relucenti@uniroma1.it; 5PhD Program in Cultural, Heritage, Education and Territory, Tor Vergata University of Rome, Via Columbia 1, 00133 Rome, Italy; 6YOCOCU, YOuth in COnservation of CUltural Heritage, Via T. Tasso 108, 00185 Rome, Italy; andrea.macchia@uniroma1.it; 7Envint S.r.l., Via Paradiso 65a, Montopoli di Sabina, 02434 Rieti, Italy; ivo.allegrini@tiscali.it

**Keywords:** aquaponics, sustainability, precision farming, iron, lettuce, micro-vectors, nanoparticles

## Abstract

Aquaponics is an innovative agricultural method combining aquaculture and hydroponics. However, this balance can lead to the gradual depletion of essential micronutrients, particularly iron. Over time, decreasing iron levels can negatively impact plant health and productivity, making the monitoring and management of iron in aquaponic systems vital. This study investigates the use of Fe-Alg-CaCO_3_ microparticles (MPs) as foliar fertilizer on lettuce plants in an aquaponic system. The research investigated *Lactuca sativa* L. cv. Foglia di Quercia Verde plants as the experimental cultivar. Three iron concentrations (10, 50, and 250 ppm) were tested, with 15 plants per treatment group, plus a control group receiving only sterile double-distilled water. The Fe-Alg-CaCO_3_ MPs and ultrapure water were applied directly to the leaves using a specialized nebulizer. Foliar nebulization was chosen for its precision and minimal resource use, aligning with the sustainability goals of aquaponic cultivation. The research evaluated rosette diameter, root length, fresh weight, soluble solids concentration, levels of photosynthetic pigments, and phenolic and flavonoid content. The 250 ppm treatment produced the most notable enhancements in both biomass yield and quality, highlighting the potential of precision fertilizers to boost sustainability and efficiency in aquaponic systems. In fact, the most significant increases involved biomass production, particularly in the edible portions, along with photosynthetic pigment levels. Additionally, the analysis of secondary metabolite content, such as phenols and flavonoids, revealed no reduction compared to the control group, meaning that the proposed fertilizer did not negatively impact the biosynthetic pathways of these bioactive compounds. This study opens new possibilities in aquaponics cultivation, highlighting the potential of precision fertilizers to enhance sustainability and productivity in soilless agriculture.

## 1. Introduction

Aquaponics is a technique that combines aquaculture with hydroponic cultivation of plant species of interest [1]. In aquaponics, a harmonious symbiotic relationship unfolds as plants absorb nutrients from fish waste, while beneficial bacteria convert ammonia into nitrates, creating a mutually supportive ecosystem in which plants flourish, fish thrive, and bacteria play a vital role in nutrient cycling [2]. This cultivation method is growing more and more, especially in urban contexts, in line with the huge demand for food resources of an ever-growing population [3]. In these terms, soilless cultivation techniques, such as aquaponics, can guarantee the filling of the gap between the increasing food demand, resource availability, and traditional cultivation systems, which are no longer sustainable [4]. Climate changes must also be considered since agriculture is the economic sector most affected by extreme weather conditions [5]. The imbalances brought about by climate change risk not only damaging a huge production system, but also perpetuating an increase in food inequality between supply and demand [6]. To date, the production of vegetables according to traditional methods on soil and the breeding of fish species suitable for their consumption are widely used in numerous regions of the planet and have a negative impact on the environment, in terms of processes connected directly or indirectly to soil erosion, pollution from pesticides and fertilizers, waste of water, and many others [7,8]. In these terms, aquaponics provides a sustainable and efficient farming technique that harnesses the mutually beneficial relationship between plants and fish to maximize yields, conserve water, and support environmentally friendly agriculture [9]. However, growing crops in aquaponic systems may encounter issues related to the depletion of essential micronutrients (e.g., Fe, Mn, Zn, B, Mo, and Cu), leading to suboptimal plant growth and requiring careful monitoring and supplementation to ensure sustainable cultivation [10,11]. The size of an aquaponic system affects the scalability of fish-to-plant production based on the availability of waste-derived nutrients. While fish nutrition is managed through feed, plant nutrition is more complex due to nutrient dynamics across the production cycle, varying plant demands, and physicochemical factors influencing nutrient bioavailability [12]. In fact, aquaponic systems that rely solely on fish waste to provide nutrients for plants, and micronutrient concentrations, in particular iron, are usually insufficient to support the successful cultivation of hydroponic vegetables [13]. This usually happens since commercial fish feeds lack iron concentration [13], because iron is also used by fish for their physiological activities, and this is reflected in the amounts of this key element in the recirculating water. Iron integration was the focus of the present study. The most common practices for adding iron in aquaponics involve either fortifying fish feed, with iron salts, or introducing iron chelates directly into the water [14,15,16]. Care must be taken with the concentration, as excessive levels can be harmful to fish and/or beneficial bacteria. The integration of modern micro- and nanotechnologies in aquaponics can ensure the overcoming of the problem related to nutrient depletion. Micro- and nanotechnology in agriculture involves using microscale and nanoscale materials and techniques to boost crop productivity, improve nutrient delivery, and enable precision farming, leading to a new era of innovative and sustainable agricultural solutions [17,18]. The controlled application of iron nanoparticles (NPs) highlights their potential to fine-tune nutrient availability within the system, contributing to the success of aquaponic cultivation [19]. In the current study, we investigated the impact of iron-functionalized calcium carbonate microparticles (MPs) on the growth and production of aquaponically grown *Lactuca sativa* L. cv. Foglia di Quercia Verde plants. Micro- or nano-scale calcium carbonate particles can serve as efficient carriers for the controlled release of nutrients that are bound to or embedded within them [20,21]. The advantage of this research project is developed on two fronts. First, the foliar administration of the smart fertilizer promotes precision agriculture, without interacting with the other components of the system; secondly, the application of micro- and nanotechnologies in the field of aquaponic cultivation is underexplored in the literature. So, can smart fertilizers, designed to target specific nutrients, effectively address the deficiencies of selective micronutrients in aquaponic systems? Furthermore, do these foliar fertilizers enhance the productivity and yield of edible plant parts, such as leaves? This groundbreaking study not only marks a new frontier but also serves as a catalyst for future exploration, providing a fresh perspective on the transformative potential of new fertilizers in aquaponics to elevate the performance of cultivated plant species.

## 2. Results

### 2.1. Scanning Electron Microscopy/Energy Dispersive X-Ray Analysis (SEM/EDX) and Fourier Transform Infrared Spectroscopy (FTIR) Characterization of Iron-Functionalized Calcium Carbonate Microspheres (Fe-Alg-CaCO_3_ MPs)

In Figure 1, MPs (Figure 1A) are between 3 and 5 µm in size and have a cavity within them (see Figure 1B). They show aggregation of smaller units with a minimum diameter of 100 nm (inset of B). EDX analysis (Figure 1C,D) revealed the presence of calcium, oxygen, and iron. The peaks for platinum and copper are due to the platinum metallization procedure and the sample support, respectively. In particular, the presence of Fe demonstrates the trapping of this element inside the MPs, which function as carriers for iron delivery.

According to the EDX evidence about Fe presence in the samples, the FTIR study clearly supports these results, showing the typical fingerprint of the Fe–O stretching modes (recorded around 563.64 cm^−1^) and the other one at 475 cm^−1^, attributed to bending vibrations of Fe–O–Fe [22], as shown in Table 1. Furthermore, the signals of carbonate and alginate (cross-linker) are also highlighted very well in the FTIR spectrum, also as shown in Table 1.

### 2.2. Bio-Morphology: Size, Length, and Fresh Weight

The growth of experimental lettuce plants demonstrated positive responses to treatments involving foliar-sprayed Fe-Alg-CaCO_3_ MPs (Figure 2). Specifically, the data pertaining to rosette diameter, root length, rosette, and root fresh weight (FW) (Figure 3) consistently exhibited increments after the application of these MPs. Specifically focusing on the rosette diameter parameter, the 50 ppm treatment exhibited a notable and statistically significant increase compared to both the CT (+39.87%) and the 10 ppm (+39.87%) treatment. Additionally, the 250 ppm treatment followed a similar trend, displaying even greater increments in comparison to both the CT (+49.02%) and the 10 ppm (+50.80%) treatment. On the other hand, the data pertaining to root length revealed a statistically significant increase solely between the 250 ppm and 10 ppm treatments (+86.74%). Significant statistical variations in the FW of both the rosette and roots were observed in the 250 ppm group compared to both the CT and the plants treated at 10 ppm. The FW of the rosette saw a percentage increase of +229.49% and +311.84%, respectively, while the FW of the roots exhibited respective increases of +131.37% and +129.89%.

### 2.3. Impact on Sugar Concentration

Refractometric analysis of soluble solids content (SSC) did not uncover substantial variations under the treatments. The measured SSC levels remained relatively stable across the experimental conditions, suggesting a consistent influence on the soluble solids content.

### 2.4. Effects on Photosynthetic Pigments, Phenols, and Flavonoids Content

The acquired data revealed that the effect of iron NPs was specifically confined to the photosynthetic pigment component (Figure 4). In specific detail, the data of chlorophyll *a* (CHL *a*) exhibited a percentage increase of +13.90% when comparing the 50 ppm treatment with the CT group. Concerning chlorophyll *b* (CHL *b*), the data exhibited notable differences among plants subjected to 250 ppm treatment, as opposed to both the CT and 10 ppm groups, revealing percentage increases of +139.80% and +100.77%, respectively. The total chlorophyll (CHL tot) content exhibited an increase in the 250 ppm group, compared with the CT (+59.41%) and 10 ppm (+54.19%) groups. No significant differences were observed in terms of carotenoids (CAR), phenols (PHE), and flavonoids (FLA) content in this experiment.

## 3. Discussion

The objective of this study was to examine the impact of Fe-Alg-CaCO_3_ MPs on the morphological characteristics and quantitative/qualitative parameters of lettuce plants cultivated under an aquaponics system. Given the extensive utilization of lettuce in aquaponics cultivation [26,27], this study employed *L. sativa*, specifically the cultivar Foglia di Quercia Verde, as the model plant. The use of the described MPs presents a potential sustainable and ecologically friendly alternative to conventional nanotechnologies for fertilization in aquaponic cultivation [19]. In the realm of soilless agriculture, this prospect holds promise for future opportunities to escalate toward large-scale production, paving the way for enhanced agricultural productivity and efficiency. The decision to incorporate iron NPs into calcium carbonate MPs was driven by the deficiency of this element in the experimental aquaponics system. The pivotal role of iron in plants, which is essential for processes, such as chlorophyll synthesis, electron transport during photosynthesis, DNA synthesis, and nitrogen reduction, underscores its significance [13]. In aquaponically grown plants, a recurring challenge manifests as the gradual depletion of iron over time, necessitating strategic supplementation to sustain optimal physiological functions and overall plant health [10]. In this investigation, we examined the effects of foliar application of iron NPs at three distinct concentrations (10 ppm, 50 ppm, and 250 ppm). As a comparative measure, the control group received a foliar spray of sterile double-distilled water. The experimental choice of the used concentrations was guided by two main factors. First, considering the limited existing literature on the use of these NPs as fertilizers for aquaponics cultivation, the concentrations were determined empirically. Second, the goal was to achieve meaningful results with relatively low concentrations, aligning with a sustainability policy. The parameters examined encompassed morphological and morphometric analyses, specifically focusing on rosette diameter, root length, rosette fresh weight, and root fresh weight. The other goals of this study were to evaluate and quantify the soluble solids content, photosynthetic pigment content, and phenols and flavonoids content, offering a comprehensive analysis of key biochemical parameters to enhance our understanding of the plant’s physiological responses under the specific experimental conditions. The noteworthy findings derived from this experiment underscored the intimate correlation between the involvement of iron in plants and their corresponding reactions, manifested through an increase in the synthesis of photosynthetic pigments. This, in turn, established a crucial link to the increased biomass observed in lettuce plants. The findings align with those of other studies, confirming that the administration of iron NPs positively influences the growth of experimental plants [28]. Specifically, there was a notable increase in the diameter of the rosette, as well as the length of the roots, particularly evident in the 250 ppm treatment. The fresh weight outcomes were consistent with the data obtained regarding dimensions. The fertilization with iron NPs positively influenced the biosynthesis of chlorophylls, thereby promoting heightened plant growth and improving physiological conditions [29]. These pigments are essential for capturing light energy during photosynthesis [30]. The products of photosynthesis, particularly glucose, serve as building blocks for plant growth and development. Glucose is used for energy, but, also, to synthesize complex molecules like cellulose, proteins, and lipids [31]. As the rate of photosynthesis increases, there is greater availability of growth resources, resulting in increased biomass, larger leaves, and overall plant development. In our detailed investigation, a more pronounced response was noted for CHL *b* than for CHL *a*. This observation can be attributed to the use of shade cloths during experimental cultivation, which were employed to mitigate the impact of elevated temperatures and intense summer sunlight. The use of shade cloths can favor the synthesis of CHL *b* over CHL *a* under certain conditions; generally, the ratio of CHL *a* to CHL *b* decreases with a decrease in irradiance, so that CHL *b* may be relatively more advantageous for plants because it absorbs light in the blue and red regions of the spectrum, complementing the filtered light from the shade [32]. The CHL tot data encapsulates the general enhancement of photosynthetic pigment biosynthesis facilitated by iron NPs. Specifically, in this study, an elevation in the CHL tot concentration was observed in plants treated with 250 ppm, surpassing both the CT and the lower concentration treatment. The elevated chlorophyll content, as revealed by treatment with iron MPs, emphasizes the central role of iron in the biosynthesis of these indispensable photosynthetic pigments. During the synthesis process, iron plays a crucial role in forming the porphyrin ring, which constitutes the core structure of chlorophyll molecules; the iron atom supports the stabilization of the chlorophyll structure, enabling its role in capturing light energy during photosynthesis [10]. Since the increase in photosynthetic activity, in terms of a greater capacity to synthesize photosynthetic pigments, is correlated with an increase in biomass, other studies in the literature report similar results to those obtained in the present study [33,34]. The absence of notable differences in CAR content across different treatments, including various concentrations of iron NPs and a sterile double-distilled water CT group in our study, indicates that carotenoid synthesis might not be directly affected by alterations in iron availability. Possible explanations include the existence of alternative pathways, compensatory mechanisms, the prioritization of iron usage for essential processes like chlorophyll synthesis, potential interactions with other nutrients, and the use of shade cloths [35,36]. SSC values serve as a significant indicator of sugar quantity, although they do not solely represent sugars. Their measurement can be utilized as a broad indicator reflecting the nutritional and physiological quality of the plant [37]. The absence of significant variation in SSC results aligns with findings from previous studies where iron was applied through nanotechnology in aquaponics; consequently, these results led us to hypothesize that the foliar application of iron NPs did not adversely impact the qualitative and physiological aspects of lettuce plants [38]. Like SSC values, the concentrations of PHE and FLA did not exhibit significant variations among the different treatments. PHE and FLA are classes of organic compounds found in plants [39]. PHE has a hydroxyl group attached to an aromatic benzene ring, contributing to functions like defense, growth regulation, pollinator attraction, and pathogenic resistance [40]. FLA, a subgroup of phenolics, has a distinctive structure with two aromatic rings linked by a three-carbon chain, serving roles in UV filtration, pigmentation, and defense mechanisms [41]. Both classes play crucial roles in plant physiology, and their concentrations can indicate the response of plants to environmental factors. The synthesis and regulation of PHE and FLA involve complex biochemical pathways. The pathways responsible for producing these compounds may not be directly influenced by the application of iron-based NPs in the concentrations used. The biosynthesis of these secondary metabolites primarily relies on enzymatic pathways, genetic regulation, and environmental factors; although iron is essential for overall plant health, its role in specific pathways leading to phenols and flavonoids is indirect [42,43]. Plants allocate resources based on their metabolic priorities. In this case, iron-based NPs may not have prompted a significant shift in resource allocation toward the biosynthesis of PHE and FLA, as other essential processes could take precedence (e.g., chlorophylls). The deployment of shade cloths might once more be implicated in influencing the biosynthetic processes leading to these metabolites, thus constraining their synthesis in preference for other compounds [44]. The protective function of the sheets could indeed have hindered the production of phenols and flavonoids precisely because the plants were not exposed to photo-induced stress or other potentially challenging stimuli [45].

## 4. Conclusions

In this study, the foliar application of Fe-Alg-CaCO_3_ MPs on aquaponically grown lettuce plants yielded insightful findings regarding its impact on various physiological and biochemical parameters. The concentration value of 250 ppm emerged as particularly noteworthy, demonstrating superior biomass production and chlorophyll concentration outcomes. This is of considerable importance in terms of both productivity and economic impact. No discernible evidence was found for significant changes in CAR, SSC, PHE, and FLA concentrations across the different treatments. These observations suggest the selective impact of iron MPs on specific biochemical pathways, indicating a nuanced and concentration-dependent response. In conclusion, although Fe-Alg-CaCO_3_ MPs demonstrated positive effects on biomass and chlorophyll concentrations, the nuanced and selective impact on other biochemical parameters necessitates further research. These findings contribute valuable insights to the field of MPs-plant interactions and lay the foundation for more targeted and comprehensive investigations in the quest for sustainable agricultural practices. Starting from these findings, further studies will be carried out to produce nanoparticles to test directly in plants.

## 5. Materials and Methods

### 5.1. Aquaponics Setup

Between March 2023 and July 2023, experiments were carried out in the aquaponics greenhouse located at the Botanical Gardens of Tor Vergata University in Rome (Rome, Italy). The aquaponics system featured two 4000 L fish tanks housing tilapias (*Oreochromis niloticus* L.). The system included a UV sterilizer, a reverse osmosis unit, a 5000 L static biofilter filled with bio-media, and a bottom-up oxygenation system. The flow rate in the biofilter was 2.5 m^3^ h^−1^, with a retention time of 0.8 h ± 45 min. This biofilter supplied nutrients to two floating raft system units, each covering a 27.5 m^2^ surface for vegetable cultivation. The tilapias were fed a diet containing 35% protein.

### 5.2. Water Quality and Environmental Parameters Monitoring

Specific sensors within the structure recorded data concerning the aquaponics system parameters, including temperature entering the raft cultivation units (T1), temperature leaving the raft cultivation units (T2), pH, and dissolved oxygen (DO) (Table 2). Abiotic data related to the greenhouse, encompassing minimum (T_min_) and maximum (T_max_) temperatures as well as minimum (RH_min_) and maximum (RH_max_) relative humidity, were documented using a thermohydrometer (Tabel 1). Weekly monitoring of the aquaponics system’s water characteristics was conducted using specialized spectrophotometric tests (Hanna Instruments, Woonsocket, RI, USA). These tests enabled the observation of the evolving concentrations of both macro- and micronutrients over time, including N (Ammonia, Nitrites, and Nitrates), P, K, Fe, Mg, S, Mo, Cl, Zn, Ca, Mn, and Cu (Table 3).

### 5.3. Synthesis of Fe-Alg-CaCO_3_ MPs

The synthesis of Fe-Alg-CaCO_3_ MPs (Figure 5) was carried out in two steps, such as the following steps:

Step 1: Synthesis of CaCO_3_ nanoparticles as a precursor of microspheres. The CaCO_3_ nanoparticles were synthetized, as reported in Valentini et al. [46], by applying an enzyme, as urease biocatalyst which, in the presence of urea and CaCl_2_, provides the CaCO_3_ nanoparticles precipitation. The urease substrate was introduced into the CaCl_2_ solution, and the enzymatic reaction products were formed within 30 min. The resulting white precipitate was rinsed with deionized water and vacuum-filtered to eliminate excess precursors. The precipitate was air-dried and subjected to both morphological and structural analyses. This synthetic approach also yields several grams of the compound, indicating potential for large-scale production [46]. For the synthesis of CaCO_3_ nanoparticles, all reagents, including anhydrous calcium chloride, urea phosphate, and urease enzyme (type III) from Jack beans (*Canavalia ensiformis* L.), were sourced from Sigma-Aldrich (Buchs, Switzerland). All mentioned reagents are of analytical grade.

Step 2: Synthesis of Fe-Alg-CaCO_3_ microspheres. The synthesis of Fe-Alg-CaCO_3_ MPs has been performed according to the literature [47], but using here, for the first time, nanostructured CaCO_3_ particles, synthetized as reported in Valentini et al. [46]. Briefly, the subsequent fabrication of iron-functionalized CaCO_3_ microspheres was carried out by pacing 400 mg of CaCO_3_ nano-powder into 100 mL of 2% (*w/v*) of sodium alginate (Alg) and leaving for intensive agitation (10 min at room temperature) in a shaker producing sodium alginate containing CaCO_3_ micro-particles (Alg-CaCO_3_ MPs). Then, Alg-CaCO_3_ particles were precipitated via centrifugation (16,128× *g*, 10 min) and washed in pure water. The washing procedure was repeated 3 consecutive times. Further injection of 100 mL of 0.1 M FeCl_2_ (1000 mg) to 100 mL of Alg-CaCO_3_ promoted the cross-linking of sodium alginate. The mixture was placed on a magnetic plate for stirring (5 min) in a shaker and then separated by centrifugation (working at 1008× *g*, for 3 min) and washed with pure water. Formed microspheres incorporating Fe element (to obtain functionalized Fe-Alg-CaCO_3_ MPs, as shown also in Figure 1, were collected by centrifugation (1008× *g*, for 3 min) and thoroughly washed. These prepared microspheres were stored in distilled water at 4 °C, until use. All chemicals, including sodium alginate, and iron dichloride (FeCl_2_), were purchased from Sigma-Aldrich, and they are of analytical grade. In all experiments, ultrapure water was prepared by using a Milli-Q^®^ IQ 7000/03/05/10/15 system.

### 5.4. Characterization of Fe-Alg-CaCO_3_ MPs

New microspheres were characterized from a morphological point of view, by applying SEM/EDX. For the sample preparation protocol for SEM, a small amount of microparticles was suspended in 100% ethanol in a glass tube and sonicated for 1 h. As the sonication ended, 10 µL of the suspension was immediately deposited on 200 mesh Formvar film Cu grids (Ted Pella, Redding, CA, USA) by drop-casting. The grid was placed on absorbent paper in a Petri dish, covered, and dried at room temperature. When the grid was perfectly dry, it was mounted on an aluminum stub by carbon tape and placed into a sputter-coated (sputter coater model K550, EMitech, Corato, Italy) and coated with platinum at 15 mA for 1.5 min. Samples were observed using a VP-SEM, Hitachi SU3500 scanning electron microscope (Hitachi, Tokyo, Japan) operated at 15–20 kV under high vacuum conditions [48,49,50]. The scanning electron microscope used in this study was equipped with a dual-energy dispersive X-ray spectroscopy detector (dEDS, Bruker XFlash^®^ 6|60, Billerica, MA, USA). This instrument can simultaneously perform multimodal imaging and spatial distribution chemical mapping, providing a truly powerful analytical approach for studying biological surfaces in their native state. The XFlash^®^ 6|60 is particularly suitable for applications with relatively low X-ray yields, as is common in nano analysis [43,44]. SEM images were analyzed using the Hitachi Map 3D 8.2 Digital Surf software (Besançon, France) [51,52]. To further characterize the MPs, FTIR was performed in transmittance mode on samples assembled in KBr pellets using a Shimadzu Model Prestige 21 spectrophotometer model apparatus, according to our previous paper [53,54].

### 5.5. Botanical Specimens and Trial Configuration

In the experimental design, we utilized floating panels made of closed-cell expanded polystyrene, which were perforated to accommodate PET growth baskets. An inert substrate of approximately 50 g of red lapillus was used, providing robust support for cultivated plants due to its grain size. Ten *Lactuca sativa* L. cv. Foglia di Quercia Verde seeds (Blumen Vegetal Seeds, Piacenza, Italy) were directly sown on the substrate surface. After a ten-day period, the recently germinated plants were systematically chosen to achieve homogeneity in the internal experiments, with only one plant retained per growth basket. Each treatment comprised three floating panels, each hosting five lettuce plants. The treatments included foliar spraying of iron NPs at varying concentrations (10 ppm, 50 ppm, and 250 ppm), alongside a control (CT) group in which plants were treated with sterile double-distilled water.

### 5.6. Fe-Alg-CaCO_3_ MPs Administrations

At the 21st day post-sowing mark, the application of Fe-Alg-CaCO_3_ MPs treatments was initiated, with administration conducted every seven days until the harvesting of plant material at the 55th day post-sowing point. This process involved a total of five applications. Specifically, the initial application utilized a volume of 1.5 mL of the product per plant, followed by 2 mL for the second administration, 3 mL for the third, 4 mL for the fourth, and a final 10 mL for the fifth, all administered through a foliar spray. The volumes required for experimental purposes were measured using Gilson^®^ pipettes along with the corresponding tips for precise and accurate dispensing. The selection of the sprayed volume in the five administrations involved tests to ensure complete coverage of the leaf surface, including both the lower and upper lamina, at each specific stage of growth. To prevent experimental contamination between treatments, especially when different treatments involved adjacent floating panels, a mobile plexiglass panel was employed. Exclusively targeting experimental plants through foliar application presents a notable advantage in aquaponics systems, because it mitigates the risk of adverse effects on raised fish and beneficial bacteria within the system.

### 5.7. Sampling Plants, Assessing Biomass, and Measuring Size

After collecting the plants, certain samples underwent immediate analysis to assess both length and fresh weight measurements, while the remaining samples were stored at −80 °C for subsequent investigations. The length-related data encompassed both rosette diameter and root length. Length analyses were conducted by placing a Canon EOS 550D camera at a height of 70 cm from the samples and capturing photos. Afterwards, these images were processed using ImageJ 1.8.0 software (U.S. National Institution of Health, Bethesda, MD, USA) capable of accurately determining the relative lengths. FW was assessed using a precision scale and involved both rosette and root specimen data.

### 5.8. Refractometric Assays: Soluble Solids Content Evaluation

For the SSC analysis, the protocol outlined by Braglia et al. was followed [55]. In summary, a combined 1.5 g of FW from randomly selected samples representing inner, middle, and outer leaves was homogenized using a mortar and pestle in liquid nitrogen. The homogenate was then centrifuged at 6089× *g* for 10 min. Subsequently, 100 µL of the supernatant was collected and examined using a digital refractometer (model HI96800; Hanna, Woonsocket, RI, USA). This analytical approach enabled the detection of sugar and other soluble solids content in the extracts, represented as the Brix value.

### 5.9. Photosynthetic Pigments: Chlorophyll a, Chlorophyll b, Total Chlorophyll, and Carotenoids Content

A combined 500 mg of FW from randomly selected samples representing inner, middle, and outer leaves was homogenized with liquid nitrogen using a mortar and pestle. Pigments were then extracted using 1.5 mL of 80% acetone in a controlled environment at 4 °C, allowing for extraction over 24 h in the dark. Subsequently, the extract was centrifuged at 4500× *g* for 10 min, and the supernatant was collected for pigment determination. CHL *a* and CHL *b*, as well as carotenoid CAR content, were determined using equations previously described by Lichtenthaler [56]. The absorbance of the extracts was measured at 663, 644, and 452 nm using a spectrophotometer (model Iris HI801; Hanna, Woonsocket, RI, USA). CHL tot was calculated as the combined sum of CHL *a* and CHL *b*. Results were expressed as µg g^−1^ of the FW.

### 5.10. Total Phenol and Flavonoid Content

The two methods followed specific protocols to measure the contents of PHE [57] and FLA [58]. A total of 500 mg of FW from randomly selected samples of the rosettes’ inner, middle, and outer leaves was homogenized using a mortar and pestle with liquid nitrogen. The resulting homogenate was then extracted overnight with 1.5 mL of pure methanol for PHE analysis and with 1.5 mL of 50% methanol for FLA analysis. The extraction process was carried out on an orbital shaker at 110 rpm at room temperature for 48 h. Afterward, the plant samples were centrifuged at 8603× *g* for 20 min, and the supernatants were collected. For PHE determination, a 200 µL aliquot of the extract was mixed with 1 mL of Folin–Ciocalteu reagent (diluted 1:10; *v/v*) and 800 µL of 1 M Na_2_CO_3_. This mixture was incubated at room temperature for 1 h. For FLA quantification, the aluminum chloride method was applied. Here, 200 µL of plant extract was combined with 40 µL of 10% AlCl_3_, 40 µL of 1 M CH_3_CO_2_K, 600 µL of methanol, and 1120 µL of distilled water. The reaction mixture was allowed to sit at room temperature for 30 min. The spectrophotometric measurements (model Iris HI801; Hanna, Woonsocket, RI, USA) for PHE and FLA were performed by recording the absorbance at 765 nm and 415 nm, respectively. These readings were then quantified using calibration curves created with increasing amounts of gallic acid (GA) as the standard equivalent (E) for phenols and quercetin (Q) for flavonoids. The results were expressed as µg GAE g^−1^ of FW for phenols and µg QE g^−1^ of FW for flavonoids.

### 5.11. Statistical Analyses

Measurements were conducted in triplicates, and the results are expressed as mean ± SD (standard deviation) values. Statistical analyses were performed using GraphPad Prism 10.1.2 software (GraphPad Software Inc., San Diego, CA, USA). Shapiro–Wilk tests were used to assess the normal distribution of all analyzed data. The findings indicated that distributions tended toward normality (*p* < 0.05), enabling the application of parametric tests (ANOVA) for data investigation and comparison. Furthermore, the parametric Tukey’s post hoc test was employed. Statistically significant differences were considered at *p* < 0.05. Comparative analyses were performed between the micro-fertilization treatments and the CT group, as well as among the various iron NPs administration groups.

## Figures and Tables

**Figure 1 plants-13-03416-f001:**
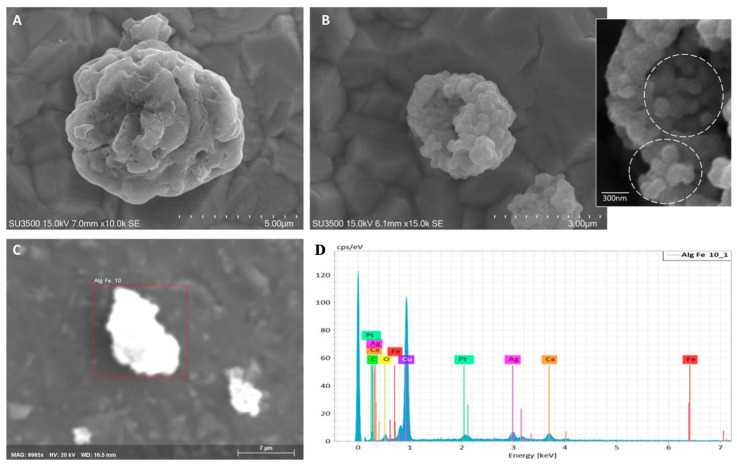
(**A**) SEM magnification 10 K: a microsphere is illustrated, it has a rough surface, due to incomplete fusion of constituent subunits. (**B**) SEM magnification 10 K: this image shows the microsphere inner cavity; the surface is roughest than (**A**), and constituent subunits are well visible; they have a minimum diameter of 100 nm, inset. (**C**) Region of interest (ROI) for EDX analysis. (**D**) EDX analysis element graph shows the presence of calcium, oxygen, and a small amount of Fe. Platinum, copper, and silver peaks are due to the platinum coating, the copper grid where the sample is placed, and the aluminum supporting stub.

**Figure 2 plants-13-03416-f002:**
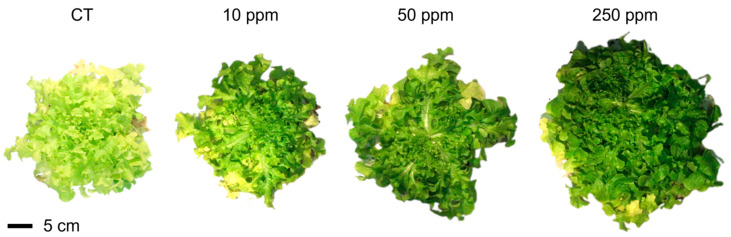
Plant samples collected on the 55th day from sowing at the end of each Fe-Alg-CaCO_3_ MPs treatment (CT, 10, 50, and 250 ppm).

**Figure 3 plants-13-03416-f003:**
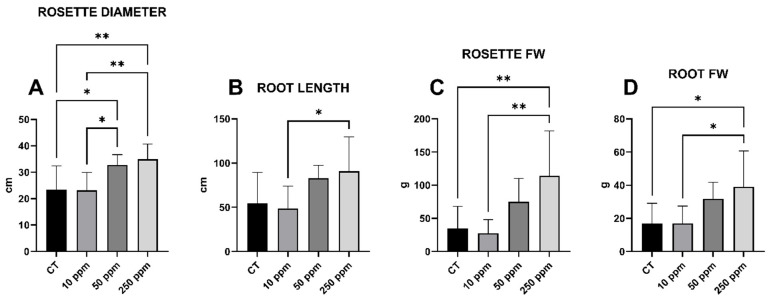
Morpho-biometrical parameters. In detail, (**A**) rosette diameter; (**B**) root length; (**C**) rosette fresh weight; (**D**) root fresh weight. The *x*-axis denotes the treatments, while the *y*-axis represents the units of measurement. The significance resulting from the comparisons between the various treatments is indicated by asterisks: * *p* < 0.05; ** *p* < 0.005.

**Figure 4 plants-13-03416-f004:**
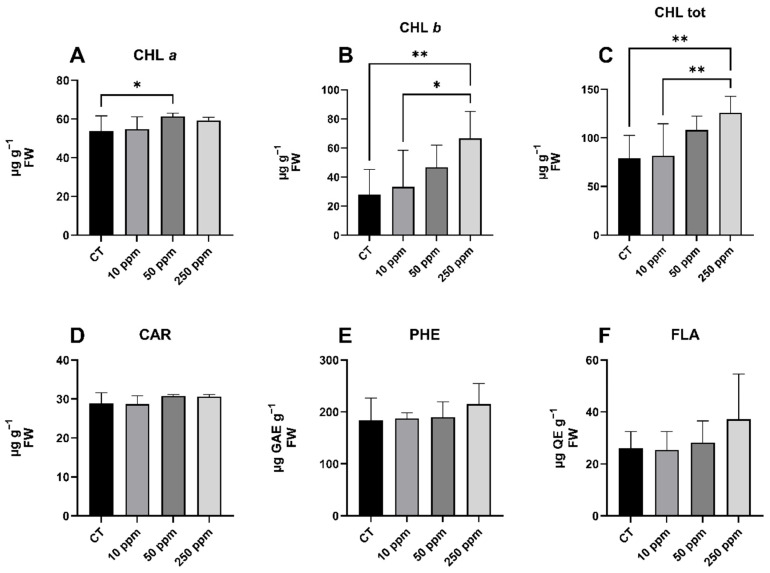
Qualitative and quantitative data from spectrophotometric assays. In detail, (**A**) chlorophyll *a*; (**B**) chlorophyll *b*; (**C**) total chlorophyll; (**D**) carotenoids; (**E**) total phenolic content; (**F**) total flavonoid content. The *x*-axis denotes the treatments, and the *y*-axis represents units of measurement. The significance resulting from the comparisons between the various treatments is indicated by asterisks: * *p* < 0.05; ** *p* < 0.005.

**Figure 5 plants-13-03416-f005:**
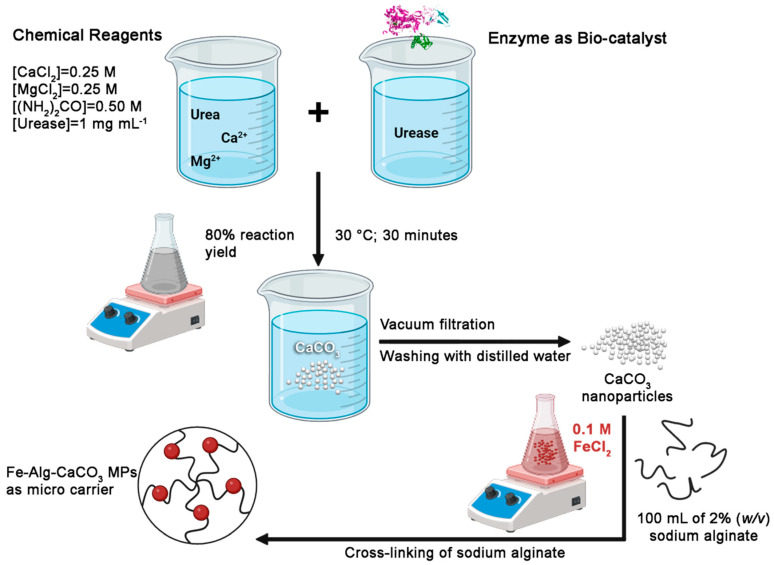
Representative flow chart for the biomineralization synthetic approach able to produce CaCO_3_ NPs (i.e., the chemical precursor) for the second step to obtain functionalized Fe-Alg-CaCO_3_ MPs, which can be able to act as micro-carriers for plant nutrients. Created with BioRender.com.

**Table 1 plants-13-03416-t001:** FTIR molecular band assignments for the samples prepared in this work.

Sample Description	Wavenumber of the Vibrational Modes (cm^−1^)	Chemical Functional Groups	References
Alg	1027.77 ν	elongation of C–O groups	[23]
	1298.51 ν	C–O stretching vibration	
	1414.55 ν	symmetric C=O vibration	
	1612.22 ν	asymmetric stretching vibration of COO groups	
	2155.37 ν	symmetric C=O vibration	
	2925.95 ν	symmetric C=O vibration	
	3446.23 ν	O–H stretching vibrations	
Fe-Alg-CaCO_3_	475 δ	bending vibrations of Fe–O–Fe	[22,24,25]
	563.64 ν	Fe–O stretching modes	
	475 δ	bending vibrations of Fe–O–Fe	
	712 δ	bending in-plane deformation mode vibrations of the O–C–O	
	871.68 δ	out-of-plane bending vibration of the (CO_3_)^2−^	
	1054.88 ν	ν1 mode of vaterite (symmetric C–O stretching	
	1423.47 ν	asymmetric stretching (ʋ3) of C–O bond	
	1604.96 ν	symmetric stretching of carboxyl group (C(=O)OH)	
	1805 ν	ν1 + ν4 sym. stretching (CO_3_)^2−^	
	2960.25 ν	symmetric C=O vibration	
	3417.29 ν	O–H stretching vibrations	

**Table 2 plants-13-03416-t002:** Parameters evaluated in the aquaponics system. Water characteristics, temperature entering the raft cultivation units (T1), temperature leaving the raft cultivation units (T2), pH, and dissolved oxygen (DO), are shown above. The greenhouse environmental data, minimum temperature (T_min_), maximum temperature (T_max_), minimum relative humidity (RH_min_), and maximum relative humidity (RH_max_), are displayed below. The relevant parameters are accompanied by their corresponding units of measurement, which are indicated in brackets. Results are represented by their mean ± standard deviation (SD) values.

Water Characteristics			
T1 (°C)	T2 (°C)	pH	DO (mg L^−1^)
26.81 ± 2.49	26.32 ± 2.60	8.65 ± 0.17	6.85 ± 0.80
**Greenhouse Environmental Data**			
T_min_ (°C)	T_max_ (°C)	Rh_min_ (%)	Rh_max_ (%)
16.22 ± 7.62	34.83 ± 3.89	30.54 ± 14.43	85.85 ± 7.03

**Table 3 plants-13-03416-t003:** Levels of macro- and micronutrients present in the circulating water within the aquaponics system. The data were obtained using spectrophotometric analyses. The targets, with the related units of measurement indicated in brackets, the recorded values, and the detection methods are shown. Results are reported as the mean ± standard deviation (SD) values. The abbreviations used for the methods column are as follows: Eicosapentaenoic Acid (EPA), 2,4,6-Tripyridyl-S-Triazine (TPTZ), and 1-(2-Pyridylazo)-2-Naphthol (PAN).

Target	Value	Analytical Method
Ammonia (mg L^−1^)	0.41 ± 0.03	Adaptation of Nessler method D1426 (ASTM Manual of Water and Environmental Technology)
Nitrites (mg L^−1^)	0.15 ± 0.22	Adaptation of EPA 354.1 denitrogenating method
Nitrates (mg L^−1^)	27.30 ± 8.11	Chromotropic acid method
Phosphorus (mg L^−1^)	3.90 ± 2.35	Adaptation of EPA 365.2 and ascorbic acid 4500-PE
Potassium (mg L^−1^)	98.25 ± 11.93	Adaptation of the turbidimetric tetraphenylborate
Iron (mg L^−1^)	0.07 ± 0.08	Adaptation of TPTZ
Magnesium (mg L^−1^)	18.00 ± 4.76	Adaptation of calmagite method
Sulfur (mg L^−1^)	50.25 ± 9.32	Precipitation method with barium salt crystals
Molybdenum (mg L^−1^)	0.05 ± 0.10	Adaptation of the mercaptoacetic acid method
Chlorine (mg L^−1^)	16.70 ± 8.21	Adaptation of mercury (II) thiocyanate method
Zinc (mg L^−1^)	0.04 ± 0.03	Adaptation of zinc from the standard methods for the examination of water and wastewater
Calcium (mg L^−1^)	130.75 ± 51.85	Adaptation of oxalate method
Manganese (µg L^−1^)	9.75 ± 5.85	Adaptation of PAN method
Copper (µg L^−1^)	22.75 ± 8.18	Adaptation of EPA method

## Data Availability

The data underlying this article will be shared upon reasonable request to the corresponding author.

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
