# Peer review of "Enhancing Lettuce (Lactuca sativa) Productivity: Foliar Sprayed Fe-Alg-CaCO3 MPs as Fertilizers for Aquaponics Cultivation"

_plants, 2024, doi:10.3390/plants13233416_

Round 1

Reviewer 1 Report

Comments and Suggestions for Authors

In the introduction, I suggest giving more weight to the background of micronutrient deficiencies in aquaponic systems and calcium carbonate microparticles with iron and developing a more precise hypothesis.

Author Response

Comment 1: In the introduction, I suggest giving more weight to the background of micronutrient deficiencies in aquaponic systems and calcium carbonate microparticles with iron and developing a more precise hypothesis.

Response 1: Thank you for pointing this out. We agree with this comment. Therefore, we have implemented the introduction. All changes are highlighted in yellow (Chapter 1-Introduction).

Reviewer 2 Report

Comments and Suggestions for Authors

In this manuscript, the author introduce a system aquaponics and how to due with the Fe reduce in this system. This study reported useful data and could be of great value for relevant topics to the scientific community. However, i think there are several problems with this manuscript, and my main comments are as follows:

1. in the manuscript, there should have some result dates or describes, please add.

2.the significant showed in figure use different letter was more better than asterisk.

3.the discussion section contains repletion from the results-section to a very large extent. All this repetition could be deleted. You can refer to the results by writing this type of sentences: The larger x in treatment y was similar to results presented by z and could be explained by q.

Author Response

Comment 1: in the manuscript, there should have some result dates or describes, please add.

Response 1: Thank you for this comment. Experimental results are shown in the graphs (Figures 3 and 4). We consider that writing the relative numbers (average and standard deviation) for each evaluated parameter is kind of a repetition. Since this, we think that graphs are well representative, and values are clear.

Comment 2: the significant showed in figure use different letter was more better than asterisk.

Response 2: Thank you for the comment. In our opinion, changing the asterisks with letters it’s an arbitrary choice and it does not bring to a misunderstanding of the comparisons between treatments. Letters could be useful if there is too much overlap of asterisks, but we do not believe this is the case.

Comment 3: the discussion section contains repletion from the results-section to a very large extent. All this repetition could be deleted. You can refer to the results by writing this type of sentences: The larger x in treatment y was similar to results presented by z and could be explained by q.

Response 3: Thank you for this comment. We agree with this, therefore we added a couple of similar studies that support our results regarding chlorophylls content and bigger biomass (highlighted in pink). The CAR, PHE, FLA and SSC data are already in the format-type that you suggested.

Reviewer 3 Report

Comments and Suggestions for Authors

As a close loop system, the combining’s of aquaculture with hydroponics gains popularity in modern agriculture as suitable practice of food production especially in areas with limited arable lad or water resources.  Here, author showed that exogenous supplementation of the synthesized Fe-Alg-CaCO3 MPs in aquaponic systems significantly improved the Lettuce Cultivation. This study is quite interesting while following issue need to address for further consideration.

-The title looks like a title of review article descriptive, Author suggested change the title.

-The abstract is mostly discussed about the techniques of experiments, author suggested to improved the abstract by writing the findings of experiment instead of methods as materials and methods section has the details.

-although the introduction is well informative while missing the possible causes of depletion of iron availability and even in the discussion part. Author suggested to provide the logical explanation of causes of iron depilation in aquaponics, and same comments for the discussion section.

In results I miss how the exogenous Fe-Alg-CaCO3 MPs behave with other nutrients, is there any other synergistic or antagonistic relationship or not?

In section 5.6, there is missing the information about the concentration of the Fe-Alg-CaCO3 MPs, which is very important. Moreover, why author sprayed Fe-Alg-CaCO3 MPs every seven days, needs logical expiations in that section, as micronutrient need very less amount for successful crop growth.

In section 5.2 author suggested to provide information which water (tap, river or supply) used for the experiments as pH seems high (8.6).

In methodology section 5.9 author suggested to avoid the heading, subheading is enough to explain the details.

Author Response

Comment 1: The title looks like a title of review article descriptive, Author suggested change the title.

Response 1: Thank you for pointing this out. We agree with the comment and we changed the title. All changes are highlighted in green (Title section, 2-3).

Comment 2: The abstract is mostly discussed about the techniques of experiments, author suggested to improved the abstract by writing the findings of experiment instead of methods as materials and methods section has the details.

Response 2: Thank you, we agree with this comment. Since this suggestion, we modified the abstract with the integration of the general outcomes from our study. All changes are highlighted in green (Abstract section, 29-30 and 38-42)

Comment 3: although the introduction is well informative while missing the possible causes of depletion of iron availability and even in the discussion part. Author suggested to provide the logical explanation of causes of iron depilation in aquaponics, and same comments for the discussion section.

Response 3: thank you for pointing this out. We agree with this comment. Therefore, since another reviewer asked about the same topic, we modified the introduction. Changes were highlighted in yellow (Chapter 1-Introduction, 69, 71-84, 93-95, 99-102).

Comment 4: In results I miss how the exogenous Fe-Alg-CaCO3 MPs behave with other nutrients, is there any other synergistic or antagonistic relationship or not?

Response 4: Thank you for pointing this out. Micronutrients work together synergistically to support plant physiology. A deficiency in any one of them causes the plant to experience stress, disrupting its metabolism even if the other micronutrients are sufficient. Once the missing micronutrient is replenished (like iron in our case), their synergistic function is restored, allowing the plant's metabolism to return to normal. On the other hand, if the comment refers to a possible interaction with the other nutrients in the recirculating water this is not possible since we administered the Fe-Alg-CaCO3 MPs via foliar spray application.

Comment 5: In section 5.6, there is missing the information about the concentration of the Fe-Alg-CaCO3 MPs, which is very important. Moreover, why author sprayed Fe-Alg-CaCO3 MPs every seven days, needs logical expiations in that section, as micronutrient need very less amount for successful crop growth.

Response 5: Thank you for this comment. The concentrations are reported just above, in the 5.5 chapter (Botanical Specimens and Trial Configuration, 391-393). We have decided not to repeat this again. For the second part of the comment, this is a very good point. Optimal levels of iron in recirculating water for aquaponic systems are 1.5-2 g/L. In our case, the highest concentration was 250 ppm (that corresponds 250 mg/L) tested by foliar applications on cultivated plants leaves. The difference lies precisely here: the iron present in water is always available to plants grown using hydroponic methods, whereas the foliar applications conducted in our study gradually lead to the utilization of the supplied iron. We chose to administer iron every 7 days, adjusting the frequency to our specific case based on insights gained from experimental trials.

Comment 6: In section 5.2 author suggested to provide information which water (tap, river or supply) used for the experiments as pH seems high (8.6).

Response 6: Thank you for this comment. The water used in our aquaponics system comes from a well, which is why the pH is high. However, a pH between 7 and 8.5 promotes the conversion of ammonia to nitrates in the biofilter. The pH recorded during our testing is very close to the threshold and did not create problems relating to each of the components (fish, bacteria and plants).

Comment 7: In methodology section 5.9 author suggested to avoid the heading, subheading is enough to explain the details.

Response 7: Thank you for pointing this out. We agree with the comment. Therefore, we delated the heading and added the model of the spectrophotometer in 5.9 and 5.10 sections that was reported in the old heading. All changes are highlighted in green (Section 5.9, 427-428 and 436-437; Section 5.10, 439 and 453; Section 5.11, 459).

Round 2

Reviewer 3 Report

Comments and Suggestions for Authors

Author showed an effort to address the reviewers comments and would be acceptable for publication